# Low-Mineral Water Diminishes the Bone Benefits of Boron

**DOI:** 10.3390/nu16172881

**Published:** 2024-08-28

**Authors:** Ting Huang, Yuhui Hao, Yao Tan, Qijie Dai, Weiyan Chen, Ke Cui, Jiaohua Luo, Hui Zeng, Weiqun Shu, Yujing Huang

**Affiliations:** 1Department of Environmental Hygiene, College of Preventive Medicine, Army Medical University, Chongqing 400038, China; huangting08123@163.com (T.H.); xiaoyue7122@tmmu.edu.cn (Y.T.); weiyanchen@tmmu.edu.cn (W.C.); cuike@tmmu.edu.cn (K.C.); ljh978@tmmu.edu.cn (J.L.); zenghui@tmmu.edu.cn (H.Z.); 2State Key Laboratory of Trauma and Chemical Poisoning, Institute of Combined Injury, Chongqing Engineering Research Center for Nanomedicine, College of Preventive Medicine, Army Medical University, Chongqing 400038, China; yuhuihao@tmmu.edu.cn; 3Department of Orthopedics, Southwest Hospital, Army Medical University, Chongqing 400038, China; mapaler@163.com

**Keywords:** desalinated seawater, boron, low mineral water, mineral metabolism, bone microstructure, bone biomechanics

## Abstract

This study looked at how desalinated seawater, which has low minerals and high boron, could affect bone health. Prior research suggests that low mineral water may harm bone health and boron could be beneficial, but the overall impact on bone health is still unclear. Eighty-nine-week-old male Balb/C mice were allocated into eight groups and administered either tap water or purified water with varying boron concentrations (0, 5, 40, and 200 mg/L). They were kept in an environment mimicking tropical conditions (35–40 °C, 70–80% humidity) and underwent daily treadmill exercise for 13 weeks. At the 14th week, serum, femora, and lumbar vertebrae were collected for mineral metabolism, bone biomarker, microstructure, and biomechanics evaluation. Boron exposure improved bone formation, microstructure, and biomechanics initially but the benefits weakened with higher levels of exposure (*p* < 0.05). Co-exposure to purified water elevated serum boron but weakened the promotion of boron on bone minerals and the bone benefits of boron compared to tap water (*p* < 0.05). Thus, when studying the health effects of boron in desalinated seawater, it is crucial to look at various health effects beyond bone health. Furthermore, it is important to consider the mineral composition of drinking water when using boron for bone health benefits.

## 1. Introduction

Desalinated seawater is a crucial solution to the lack of traditional freshwater resources, which is an important problem faced by China as well as many other parts of the world [1]. Notably, there has been a notable uptick in the implementation of seawater desalination projects along China’s coastal regions in recent years, particularly in the production of domestic water [2,3]. As the demand for desalinated seawater rises, the associated health risks are gaining significance. Research in the Persian Gulf region has revealed that desalinated seawater used for drinking purposes frequently exhibits unfavorable sensory attributes such as taste, odor, or trigeminal nerve paresthesia [4]. The prolonged consumption of such water can result in electrolyte imbalances and elevated incidences of digestive system cancer [5]. Studies conducted in Israel have indicated that the consumption of desalinated seawater may lead to a reduction in blood magnesium levels, potentially contributing to a higher incidence of cardiovascular disease and a lower survival rate among patients with acute myocardial infarction [6]. Additionally, Zhang et al. reported that the consumption of desalinated seawater is positively associated with chronic conditions such as hypertension, diabetes, and tumors [7]. These studies not only indicate the potential health risks of desalinated seawater but also demonstrate their complexity and non-specificity.

The potential health hazards associated with desalinated seawater may be attributed to its water quality characteristics. Presently, reverse osmosis stands as the primary technology utilized for seawater desalination [2]. It imparts two distinct characteristics to the desalinated seawater. Initially, it has high boron content. Our previous study showed that the boron in desalinated seawater ranges from 0.95 mg/L to 1.97 mg/L in Hainan, China. Other studies showed that the boron in desalinated seawater ranges from 0.72 mg/L to 1.6 mg/L and 0.57 ± 0.53 mg/L in the Shengsi Archipelago and Zhoushan, China, respectively [8,9]. They are approaching or surpassing the standards for drinking water quality of China (1.0 mg/L) but much lower than the lifetime health advisory (6 mg/L) in the Drinking Water Standards and Health Advisories Tables (2018 Edition) by EPA [10]. Boron has been shown to have various health benefits, particularly in bone health, where the boron is stored [11,12]. Boron can promote the osteogenic differentiation of bone marrow-derived mesenchymal stem cells (BMSCs) and upregulate the expression of the biomarkers of bone formation in BMSCs and MC3T3-E1 cells [13,14,15]. Boron supplementation has been found to improve bone metabolism and enhance the bone’s mechanical properties in vivo [16,17]. Some studies have indicated that the beneficial effects of boron on bone health may be associated with its regulatory influence on mineral metabolism [18,19,20]. Boron can regulate the metabolism of the major skeletal elements, calcium, and magnesium [17,21,22]. Studies have shown that boron can increase the serum 17β-estradiol and decrease urinary calcium and magnesium in menopausal women [23,24,25]. Additionally, boron can enhance the absorption of calcium, magnesium, and phosphorus [26]. But boron’s effects on mineral metabolism and bone health depend on mineral intake levels. Boron supplementation decreased feed intake when birds fed on a calcium- and phosphorus-deficient diet [20]. When calcium intake is adequate, boron supplementation can enhance bone biomechanics [16]. However, the concurrent administration of an additional calcium supplement under these conditions may attenuate the beneficial effects of boron [16]. In menopausal women, boron supplementation was found to reduce the urinary excretion of phosphorus and the percentage of dietary calcium lost in the urine under the conditions of low dietary magnesium. Conversely, when magnesium was adequately supplied, boron supplementation increased the percentage of dietary calcium excreted in the urine [23,24,25].

Another notable characteristic of desalinated seawater treated through reverse osmosis is the absence of essential mineral components such as calcium and magnesium. Our previous study showed that calcium and magnesium in desalinated seawater were lower than the detection limit, with calcium at 0.08 mg/L and magnesium at 0.01 mg/L. The total hardness (expressed as CaCO_3_) of desalinated seawater was 11.4–16.2 mg/L and 26.86 ± 12.27 mg/L in the Shengsi Archipelago and Zhoushan, China, respectively [8,9]. They were lower than the recommendation by the WHO: 20 mg/L for calcium, 10 mg/L for magnesium, and 2 to 4 mmol/L for the sum of calcium and magnesium (200 mg/L to 400 mg/L as CaCO_3_) [27]. Our previous study found drinking low-mineral water may disturb calcium metabolism, and aggressive renal net acid excretion, then reduce bone minerals and destroy bone structure and biomechanics [28,29,30]. Drinking desalinated seawater may also disrupt calcium metabolism, which may diminish the bone benefits of boron. Furthermore, deficiencies in minerals present in drinking water may facilitate the absorption of other metallic elements, thereby aggravating the risk of heavy metal exposure [31,32]. Low mineral levels in desalinated seawater may also potentially increase the bioavailability of boron, resulting in excessive internal exposure to boron. Excessive boron supplementation may negate its effects on reducing fecal calcium and phosphorus excretion [20]. Boron overexposure inhibits the activity of bone marrow stromal cells and osteoblasts, leading to bone toxicity [13,15,33]. Moreover, the present utilization of desalinated seawater extends to tropical regions, where the hot and humid climate can impact human mineral metabolism [34]. Physical labor there will further exacerbate the burden and metabolism of bone and consequently elevate the risk of bone injuries [34]. Our previous research has indicated that the consumption of low-mineral water in these regions can influence mineral metabolism during physical work [35].

Therefore, the presence of boron in desalinated seawater may not confer any beneficial effects on bone health when consumed in tropical regions. Conversely, it may pose adverse health risks to the skeletal system. This study was performed to determine the bone effects of boron in desalinated seawater on BALB/c mice.

## 2. Materials and Methods

### 2.1. Animals and Reagents

The Institutional Animal Care and Ethics Committee of Army Medical University (Chongqing, China) approved all the animal procedures (Ethical number: AMUME20226107) on 11 October 2022. All the protocols involving animal care and use strictly adhered to the NIH requirements and followed the ARRIVE guidelines.

The boric acid used in this study was procured from Boer Chemical Reagent Co., Ltd. (BLN11295-500, purity ≥ 99.5%, Shanghai, China). In total, 28.6 mg/L, 228.8 mg/L, and 1144 mg/L of boric acid were added to both purified and tap water (as 5, 40, and 200 mg/L of boron). Eighty male BALB/c mice (9-week-old, weight: 16.7 g ± 2.2 g) were procured from the Experimental Animal Center of Army Military Medical University (license: SCXK2012-0003). The mice were randomly divided into eight groups (*n* = 10 per group) and were given either tap water (calcium: 37 mg/L; magnesium: 7 mg/L; phosphorus: 0.1 mg/L; boron: 19.06 μg/L, TDS: 180 ppm–220 ppm) or purified water (calcium, magnesium, phosphorus, and boron were below the detection limit; TDS: 0 ppm–5 ppm; simulating the absence of essential mineral components in desalinated seawater), without or with various concentrations of boron. The subjects were provided with standard maintenance food (Laboratory Feeds, TMMU, China license: SCX2012-0009, calcium: 10.1 mg/g; magnesium: 2.9 mg/g; phosphorus: 6.8 mg/g; boron: 30.8 μg/g) and were housed in an environment simulating tropical conditions (12 h light/dark cycle; temperature maintained at 35–40 °C; humidity levels maintained at 70–80%).

To simulate physical load, the mice were administered treadmill exercise treatment under a running speed of 15 m/min and a slope of 5° for 15 min per day, 5 days per week, with 2 days of rest, for 13 weeks. In the 14th week, the mice were euthanized using pentobarbital sodium (Nembutal, 60 mg/kg, i.p.). The 3rd lumbar vertebrae and left femora were harvested for the evaluation of bone microstructure. After scanning, the mineral content of the left femora was evaluated. The right femora were assessed for biomechanical properties. One milliliter of blood samples was extracted from the ocular region and allowed to coagulate for one hour. A total of 400 µL–500 µL of serum was obtained through centrifugation at 1174× *g* for 10 min (Eppendorf Centrifuge 5430R, Eppendorf AG, Hamburg, Germany), and subsequently stored at −80 °C for later use in serological testing and mineral analysis as outlined below.

### 2.2. Serological Testing and Serum Mineral Content Analysis

Serum bone metabolism markers and regulators were assessed using ELISA kits obtained from Shanghai Enzyme-linked Biotechnology Co., Ltd. (Shanghai, China). The biomarkers analyzed included the *N*-terminal peptide of mouse type I procollagen (PINP) (BMS2307), C-terminal cross-linked telopeptide of type-I collagen (CTx) (BMS2308), tartrate-resistant acid phosphatase (TRACP) (BMS2310), bone alkaline phosphate (BALP) (BMS2309), osteocalcin (OC) (BMS2311), the receptor activator of nuclear factor κB ligand (RANKL) (BMS2316), osteoprotegerin (OPG) (BMS2315), calcitonin (CT) (BMS2312), parathyroid hormone (PTH) (BMS2313), and vitamin D_3_ (VD_3_) (BMS2314).

A total of 100 µL serum (only three leftover serum samples exceeded 100 µL per group) was digested overnight with nitric acid (Chuandong Chemical (Group) Co., Ltd., Chongqing, China) at 55 °C. Calcium, magnesium, phosphorus, and boron were measured using inductively coupled plasma mass spectrometry (ICP-MS, ELEMENT XR, Thermo Fisher Scientific, Bremen, Germany).

### 2.3. Bone Microstructure and Mineral Content Analysis

The experimental procedures followed the established protocols as outlined in Tan et al. [30]. Specifically, the trabecular of lumbar vertebrae and cortical of left femora were conducted using a three-dimensional microcomputed tomography (micro-CT) system (SkyScan 1276, Brucker, Kontich, Belgium) operating at 50 kVp and 200 µA. The voxel size was 6 µm for the femora and 12.5 µm for the lumbar vertebrae. The evaluation of bone microstructure in rodents adhered to the guidelines set forth by Bouxsein et al. [36]. Following scanning, NRecon (version 1.7.4.6, Bruker microCT, Kontich, Belgium) was used for data reconstruction, and Data Viewer (version 1.5.6.2, Bruker microCT, Kontich, Belgium) adjusted the angles of coronal, sagittal, and horizontal images. Then, Skyscan CT-Analyser (version 1.23.0.2, Brucker AG, Billerica, MA, USA) was used to calculate bone morphological parameters within the regions of interest (ROI), which were defined as starting from 0.2 mm to 0.6 mm below the lumbar vertebra growth plate and 3.5 mm to 5.5 mm above the femoral growth plate, respectively (Appendix A). The parameters including vertebral bone volume/total volume (BV/TV), trabecular bone number (Tb.N), trabecular separation (Tb.Sp), trabecular pattern factor (Tb.pf), structure model index (SMI), and bone mineral density (BMD), femoral BMD, and the mean of cortical bone area (Ct.Ar) were evaluated under the guidelines outlined by the American Society for Bone and Mineral Research (ASBMR) Histomorphometry Nomenclature Committee [37]. CTvox (version 3.3.1, Bruker microCT, Kontich, Belgium) is employed for the three-dimensional reconstruction analysis of the femur.

After scanning, the femur sample was oven-dried at 80 °C for 2 h and subsequently ground into a fine powder. In total, 15–25 mg powder was digested overnight with nitric acid (Chuandong Chemical (Group) Co., Ltd., Chongqing, China) at 55 °C. Calcium, magnesium, phosphorus, and boron were measured using inductively coupled plasma mass spectrometry (ICP-MS, ELEMENT XR, Thermo Fisher Scientific, Bremen, Germany).

### 2.4. Bone Biomechanical Analysis

The bone biomechanical testing of the right femora was conducted by a three-point bending test according to our previous study [30]. Briefly, the bone was immersed in a physiological saline solution to maintain its hydration before mechanical testing. Subsequently, the bone was positioned horizontally on two supporting bars with a span of 10 mm, and a central loading force was applied at the mid-diaphysis. The femur was subjected to tensile loading at a rate of 1 mm/s until fracture. The load/deformation curve was then recorded using a universal material testing machine (Instron1011, Instron Corporation, Norwood, MA, USA).

### 2.5. Statistical Analysis

All the analyses were performed in SPSS (IBM Corp. Released in 2020. IBM SPSS Statistics for Windows, version 27.0. Armonk, NY, USA: IBM Corp). Normal distribution was confirmed in all the tests performed by the one-sample Kolmogorov/Smirnov test. Two-way ANOVA procedure in general linear models was used to analyze the interaction effect of two factors, water type (tap water vs. purified water) and boron exposure level (0, 5, 40, and 200). Bonferroni was used to examine both main effects and simple effects. The data were reported as means ± SEM, with statistical significance defined as *p* < 0.05.

## 3. Results

### 3.1. Bodyweight, Diet, and Water Consumption

No significant differences existed in all the groups’ body weights, diet, and water consumption during 14 weeks. The intake of calcium, magnesium, and phosphorus from water was significantly lower in the groups consuming purified water (*p* < 0.05, Appendix A). However, there were no significant differences in the dietary intake and total intake of calcium, magnesium, and phosphorus between the groups consuming tap water and those consuming purified water (*p* ≥ 0.05, Appendix A). The ratio of calcium to magnesium intake, from both diet and water, was 3.49 in the mice drinking tap water and 3.48 in the mice drinking purified water (Appendix A). The boron intake from water and total boron intake significantly increased as the boron exposure increased in the groups exposed to 40 and 200 mg/L boron (*p* < 0.05, Appendix A). The contribution of boron from drinking water to total intake significantly increased as the boron exposure increased in all the boron exposure groups (*p* < 0.05, Appendix A).

### 3.2. Comparison of Calcium Regulatory Hormones and Bone Modeling Markers after Drinking Two Types of Water Combined with Boron

Drinking water type and boron exposure had interaction effects in regulating serum BALP, OCN, PINP, OPG/RANKL, TRACP, CTx, and CT (*p* < 0.05). Without boron co-exposure, the mice drinking purified water had lower serum BALP (Figure 1A), OCN (Figure 1B), and OPG/RANKL (Figure 1D), and higher CTx (Figure 1F) and CT (Figure 1H) compared to the mice drinking tap water (*p* < 0.05). The mice drinking purified water had higher OCN (Figure 1B) and CT (Figure 1H) in all the groups co-exposed to boron (*p* < 0.05), higher OPG/RANKL (Figure 1D) and CTx (Figure 1F) when co-exposed to 5 mg/L boron (*p* < 0.05), lower PINP co-exposed to 200 mg/L boron (*p* < 0.05, Figure 1C), and lower vitamin D_3_ when co-exposed to 5 and 200 mg/L boron (*p* < 0.05, Figure 1I) than the mice drinking tap water. The mice drinking purified water had lower TRACP when co-exposed to 5 mg/L boron but higher TRACP when co-exposed to 40 mg/L and 200 mg/L boron than the mice drinking tap water (*p* < 0.05, Figure 1E).

Within the groups that drank purified water, the serum OPG/RANKL was highest in the group exposed to 5 mg/L boron (*p* < 0.05, Figure 1D). The PTH levels were highest in the group exposed to 40 mg/L boron (*p* < 0.05, Figure 1G). The TRACP levels decreased in the group exposed to 5 mg/L boron but increased in the group exposed to 40 mg/L and 200 mg/L boron (*p* < 0.05, Figure 1E). The group exposed to 200 mg/L boron had the highest serum CTx (*p* < 0.05, Figure 1F) and lowest vitamin D_3_ levels (*p* < 0.05, Figure 1I). BALP significantly increased (*p* < 0.05, Figure 1A), and CT decreased when exposed to 5 mg/L and 40 mg/L boron (*p* < 0.05, Figure 1H). The OCN (Figure 1B) and PINP (Figure 1C) levels significantly increased in all the boron-exposed groups (*p* < 0.05).

In the groups drinking tap water, serum vitamin D_3_ increased in the group exposed to 5 mg/L boron (Figure 1I), and PTH (Figure 1G) increased in the group exposed to 40 mg/L boron (*p* < 0.05). BALP increased in the group exposed to 5 mg/L boron but decreased in the group exposed to 200 mg/L boron (*p* < 0.05, Figure 1A). OCN decreased in the group exposed to 5 mg/L and 200 mg/L boron (*p* < 0.05, Figure 1B). TRACP increased in the groups exposed to 40 mg/L and 200 mg/L boron (*p* < 0.05, Figure 1E). PINP (Figure 1C) and CTx (Figure 1F) increased, and CT (Figure 1H) decreased across all the boron exposure groups (*p* < 0.05).

### 3.3. Comparison of Bone Microstructures and Biomechanical Properties after Drinking Two Types of Water Combined with Boron

Drinking water type and boron exposure had interaction effects in regulating vertebral BV/TV, Tb.N, Tb.Sp, and femoral Ct.Ar (*p* < 0.05). The mice drinking purified water, compared to those drinking tap water, presented lower femur BMD, either with or without boron co-exposure (*p* < 0.05, Figure 2B). They also had lower femur Ct.Ar (Figure 2C), vertebral BMD (Figure 3B) and BV/TV (Figure 3C) across all the boron co-exposure groups (*p* < 0.05). The mice drinking purified water had lower Tb.N (Figure 3D), higher Tb.Pf (Figure 3E), and SMI (Figure 3F) of the vertebra in the groups co-exposed to 40 mg/L and 200 mg/L boron (*p* < 0.05), and higher Tb.Sp of the vertebra in the groups co-exposed to 200 mg/L boron (*p* < 0.05, Figure 3G) compared to those drinking tap water.

In the groups drinking purified water, the vertebral BV/TV (Figure 3C), Tb.N (Figure 3D), and BMD (Figure 3B) significantly increased when co-exposed to 40 mg/L boron (*p* < 0.05). In the groups drinking tap water, the vertebral Tb.Sp decreased in the group exposed to 40 mg/L boron (*p* < 0.05, Figure 3E). The femora Ct.Ar significantly increased in the groups exposed to 5 mg/L and 40 mg/L boron (*p* < 0.05, Figure 2C). When co-exposed to 40 mg/L and 200 mg/L boron, the vertebral Tb.N (Figure 3D) increased and Tb.Pf (Figure 3G) decreased (*p* < 0.05). The vertebral BV/TV (Figure 3C) and BMD (Figure 3B) increased across all the boron-exposed groups (*p* < 0.05).

The maximum load of the femora in the mice drinking tap water was higher than that in the mice drinking purified water when combined with 40 mg/L and 200 mg/L boron (*p* < 0.05, Figure 2D).

### 3.4. Comparison of Mineral Metabolism after Drinking Two Types of Water Combined with Boron

Except for bone boron, drinking water type and boron exposure had interaction effects in regulating serum and bone minerals (*p* < 0.05). The mice drinking purified water compared to those drinking tap water, exhibited higher serum boron and lower bone phosphate in all the groups co-exposed to boron (*p* < 0.05, Figure 4A,H), and had lower bone calcium and magnesium when co-exposed to 40 mg/L and 200 mg/L boron (*p* < 0.05, Figure 4D,F), and lower serum phosphate when co-exposed to 0 mg/L, 5 mg/L, and 40 mg/L boron (*p* < 0.05, Figure 4G). The serum magnesium in the mice drinking purified water decreased when co-exposed to 0 mg/L and 200 mg/L boron but increased when co-exposed to 5 mg/L boron (*p* < 0.05, Figure 4E), and serum calcium increased when co-exposed to 0 mg/L and 40 mg/L boron but decreased when co-exposed to 5 mg/L and 200 mg/L boron (*p* < 0.05, Figure 4C) compared to those in mice drinking tap water.

In the mice drinking purified water, bone boron significantly increased in the group exposed to 200 mg/L boron (*p* < 0.05, Figure 4B). The serum phosphate significantly increased when exposed to 5 mg/L boron (*p* < 0.05, Figure 4G) and the serum magnesium and phosphate increased when exposed to 5 mg/L and 40 mg/L boron (*p* < 0.05, Figure 4E). The serum boron increased in all the groups co-exposed to boron (*p* < 0.05, Figure 4A).

In the groups combined with tap water, the bone boron increased when co-exposed to 40 mg/L and 200 mg/L boron (*p* < 0.05, Figure 4B). The bone calcium (Figure 4D), magnesium (Figure 4F), and phosphate (Figure 4H) significantly increased when exposed to 40 mg/L boron (*p* < 0.05). The serum boron was higher in the group exposed to 200 mg/L boron (*p* < 0.05, Figure 4A). The serum calcium increased in the groups exposed to 5 mg/L and 200 mg/L boron (*p* < 0.05, Figure 4C). The serum phosphate significantly increased in the group co-exposed to 40 mg/L boron but decreased when exposed to 200 mg/L boron (*p* < 0.05, Figure 4G). The serum magnesium was lower in all the groups co-exposed to boron (*p* < 0.05, Figure 4E).

## 4. Discussion

This study showed that consuming low-mineral water promoted boron absorption but weakened boron’s benefits on bone health in high temperature and humidity combined with physical work.

The ratio of calcium to magnesium intake is important for bone due to their competitive absorption. In this study, the ratio of calcium to magnesium intake from both water and diet was 3.48 and 3.49. These values are close to the range of 2.2–3.2, which was identified as the most protective for bone health in Hibler’s study [38]. Additionally, there were no significant differences in the total intake of calcium and magnesium between the groups consuming different types of water. This suggests that the observed differences in bone and mineral metabolism, microstructure, and biomechanics between the groups drinking different water may not be attributed to the calcium/magnesium intake ratio.

Bone remodeling involves bone resorption and bone formation. During bone resorption, the degradation of type I collagen in the bone matrix releases CTx into the blood. Thus, serum CTx directly reflects bone resorption [39]. Exposure to boron was found to promote bone resorption in the mice drinking tap water. However, only the highest boron exposure (200 mg/L) significantly promoted bone resorption in the mice consuming purified water. This difference may result from drinking pure water alone can significantly increase bone resorption as shown in our previous studies [29,40], and the regulatory effects of boron on osteoclast activity may vary depending on the type of water consumed. TRACP is a marker of the activity of osteoclasts, which are responsible for bone resorption. In this study, osteoclast activity exhibited an increase in boron exposure at 40 mg/L and 200 mg/L either drinking tap water or purified water, but only decreased in the group drinking purified water when exposed to 5 mg/L boron. During the synthesis of bone matrix by type I collagen in bone formation, PINP from its propeptide is released into the blood. Therefore, serum PINP can be used to evaluate the rate of bone matrix synthesis [39]. This study showed that boron exposure enhanced bone formation, regardless of whether the mice drank purified water or tap water. However, when exposed to the highest boron (200 mg/L), the promotive effect of boron in the mice drinking purified water was notably lower than that in the mice drinking tap water. Osteoblasts are responsible for formation. Serum BALP, an extracellular enzyme produced by mature osteoblasts, is associated with bone mineralization and can reflect osteoblast activity [39]. Contrary to osteoclasts, osteoblast activity initially increased at 5 mg/L boron exposure and decreased as boron exposure increased (200 mg/L) in the mice drinking tap water. When combined with purified water, boron promoted osteoblast activity at lower concentrations (5 mg/L). The effect weakened at higher boron exposure (40 mg/L) and demolished at highest boron exposure (200 mg/L). As BALP, OCN is also secreted by osteoblasts. However, the majority of OCN produced by osteoblasts is incorporated into the bone matrix and only enters the bloodstream during bone resorption. Thus, serum OCN can reflect both osteoblast activity and bone turnover [39]. The mice drinking purified water had lower serum OCN than the mice drinking tap water without boron exposure, but its serum OCN increased and was higher than that in the mice drinking tap water when exposed to boron. Besides, the serum OCN in the mice drinking tap water decreased when exposed to boron (5 mg/L and 200 mg/L). These findings suggest that boron exerts differential effects on osteoblast activity and bone turnover in mice consuming different types of water.

This discrepancy may be attributed to the differential effects of boron on osteoblasts and osteoclasts when administered with purified water versus tap water. It has been proposed that boron modulates bone metabolism by regulating the regulator of bone formation and resorption [13,14,15,19]. RANKL is secreted by osteoblasts and osteocytes. It is a critical ligand for the receptor activator of nuclear factor kappa-Β (RANK) on osteoclasts, thereby promoting osteoclast differentiation and bone resorption upon binding. OPG is also produced by osteoblasts. It acts as a decoy receptor for RANKL, thereby inhibiting its binding with RANK and subsequently suppressing osteoclastogenesis. This study demonstrated that exposure to low doses of boron combined with purified water can elevate the OPG/RANKL ratio, thereby promoting bone formation and inhibiting osteoclast maturation. However, at high boron exposure (200 mg/L), purified water demolished the inhibitory effect of boron on CT, and reduced vitamin D_3_, resulting in suppressing calcium absorption and bone formation, and promoting bone resorption.

The variation in the impact of boron on bone metabolism, when consuming different water, may lead to differences in bone microstructure. In the mice drinking tap water, low-dose boron exposure (5 mg/L or 40 mg/L) demonstrated beneficial effects on bone structure by enhancing the vertebral BV/TV, Tb.N, and femoral Tt.Ar, while concurrently reducing the Tb.Pf, Tb.Pf, and SMI of the vertebrae. The advantageous effects on the vertebral Tb.Pf, SMI, and femoral Tt.Ar were attenuated as the boron exposure increased (200 mg/L). All of these benefits of boron were negated at some exposure doses in the mice drinking purified water. Differences in bone microstructure may result in differences in bone biomechanics. Maximum load is a critical indicator of bone strength [16,41,42,43,44]. The maximum load of the femora in the mice drinking purified water was lower than that in the mice drinking tap water when exposed to 40 mg/L and 200 mg/L. Consequently, we hypothesize that the low mineral content in drinking water may attenuate the benefits of boron on bone microstructure and biomechanics. This attenuation may be attributed to the influence of boron’s regulatory capacity on bone metabolism.

Besides bone matrix metabolism, bone mineral deposition affects bone structure and strength. Bone mineral deposition exhibits a stronger association with boron compared to bone matrix metabolism [45]. This study demonstrated that boron could increase the vertebral bone mineral density in the mice drinking tap water. But this effect of boron at some exposure doses vanished when in the mice drinking purified water. These findings suggest that combination with purified water attenuates the benefits of boron on BMD. The major minerals in bone are calcium, magnesium, and phosphate. Exposure to 40 mg/L boron markedly elevates the calcium, magnesium, and phosphate in the bone in the mice drinking tap water. However, these benefits on the deposition of minerals in the bone are nullified in the mice drinking purified water since the boron content in the under-mineralized bone was significantly lower compared to the fully mineralized bone [45]. Inadequate bone mineralization may in turn directly reduce the boron deposition in bone. The serum boron increased in the groups drinking purified water when exposed to boron (5 mg/L, 40 mg/L, and 200 mg/L), whereas the bone boron only increased when exposed to 200 mg/L boron. In contrast, in the groups drinking tap water, the blood boron only increased when exposed to 200 mg/L boron, but the bone boron increased when exposed to both 40 mg/L and 200 mg/L boron. In the mice drinking tap water, the effect of boron on the serum calcium levels appeared to fluctuate with the bone calcium. It is hypothesized that the promoting effect of boron on bone calcium deposition may be obscuring its effect on the blood calcium levels. This phenomenon was more pronounced in the magnesium metabolism in the mice drinking tap water. In them, the serum magnesium levels varied in response to boron’s regulation of the bone magnesium. However, drinking purified water diminished the promoting effect of boron on the bone and serum calcium. Besides, in the mice drinking purified water, although exposed to 5 mg/L and 40 mg/L boron increased the serum magnesium. The bone magnesium did not change with boron exposure. It is hypothesized that drinking purified water might diminish the regulatory impact of boron on the depositing of calcium and magnesium in bone tissue. This phenomenon may be attributed to the disruptive impact of drinking purified water on mineral metabolism [29,30]. Moreover, calcium and magnesium present in the bone matrix not only influence boron accumulation but also play a regulatory role in bone metabolism in turn [46,47]. In contrast to calcium and magnesium, boron exhibited a promotive effect on both the serum phosphorus and bone phosphorus in the groups drinking tap water (exposed to 40 mg/L boron), and on the serum phosphorus in the groups drinking purified water (exposed to 5 mg/L boron) as Ri et al. report [48]. However, with increasing boron exposure, the effect on the serum phosphorus diminished in the groups drinking purified water when exposed to 40 mg/L and 200 mg/L boron. In the mice drinking tap water, when exposed to 200 mg/L boron, the effect on the bone phosphorus diminished and serum phosphorus even decreased. This could result from higher urinary phosphorus excretion caused by excessive boron exposure [49]. The findings indicate that boron’s capacity to enhance the absorption of calcium, magnesium, and phosphorus may be less pronounced compared to its ability to facilitate the deposition of these minerals in bone tissue. Insufficient minerals in drinking water can impede this capacity of boron, and then weaken the benefits of boron on bone metabolism, microstructure, and biomechanics. High boron in drinking water will also diminish its benefits on mineral metabolism and bone.

This study also has several limitations. Firstly, to align with actual environments where desalinated seawater is utilized, we simulated environments characterized by high humidity, elevated temperatures, and a specified exercise load. However, it remains uncertain whether the influence of this heightened metabolic state on the skeletal system will obscure the real effects of purified water on the bone benefits of boron. Secondly, our concurrent epidemiological study indicated that the dietary patterns of individuals residing in regions where desalinated seawater is consumed are marked by a high caloric intake coupled with low calcium consumption. Such dietary patterns may exacerbate the effects of low-mineral water on mineral metabolism [30]. Further research is required to ascertain whether this could additionally constrain the potential bone benefits of boron in populations consuming desalinated seawater. Thirdly, the alterations in the regulatory factors of minerals and bone metabolism, such as calcitonin, parathyroid hormone, and vitamin D_3_, were inadequate to account for the observed differences in boron’s regulatory effects on mineral and bone metabolism under the two distinct drinking water conditions. Consequently, further research is warranted to elucidate the underlying mechanisms.

## 5. Conclusions

This study corroborates that a reduced mineral content in drinking water facilitates the absorption of boron, primarily manifested as an increase in blood boron levels with minimal impact on bone boron. Nevertheless, diminished mineral content in drinking water markedly impairs boron’s regulatory influence on the metabolism of calcium, magnesium, and phosphorus, as well as its bone benefits. Therefore, when evaluating the health effects of boron in desalinated seawater, it is prudent to consider a broader spectrum of health outcomes, including potential reproductive toxicity, rather than concentrating exclusively on its bone benefits. Furthermore, when utilizing boron to enhance bone health, the mineral content of drinking water should be paid attention.

## Figures and Tables

**Figure 1 nutrients-16-02881-f001:**
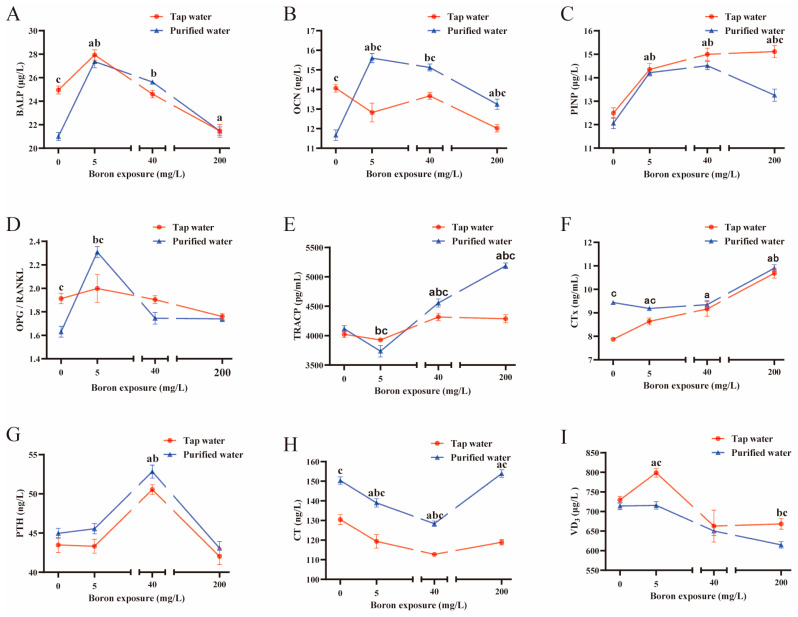
Effects of drinking two types of water with boron on serum biomarkers and regulators bone remodeling (**A**–**F**) and calcium regulatory hormones (**G**–**I**) of mice. (**A**) Serum bone alkaline phosphate concentration. (**B**) Osteocalcin concentration. (**C**) Procollagen I *N*-terminal propeptide concentration. (**D**) The ratio osteoprotegerin and receptor activator of nuclear factor-κB ligand. (**E**) Tartrate-resistant acid phosphatase concentration. (**F**) The *C*-terminal peptide of type I collagen concentration. (**G**) Parathyroid hormone concentration. (**H**) Calcitonin concentration. (**I**) Vitamin D_3_ concentration. The values are presented as means with the error bars indicating SEM; *n* = 8 mice/group. ^a^: there is a significant difference compared with the group drinking tap water without boron exposure (*p* < 0.05). ^b^: there is a significant difference compared with the group drinking purified water without boron exposure (*p* < 0.05). ^c^: there is a significant difference between the two groups which had the same exposed dose of boron but combined with different water (*p* < 0.05).

**Figure 2 nutrients-16-02881-f002:**
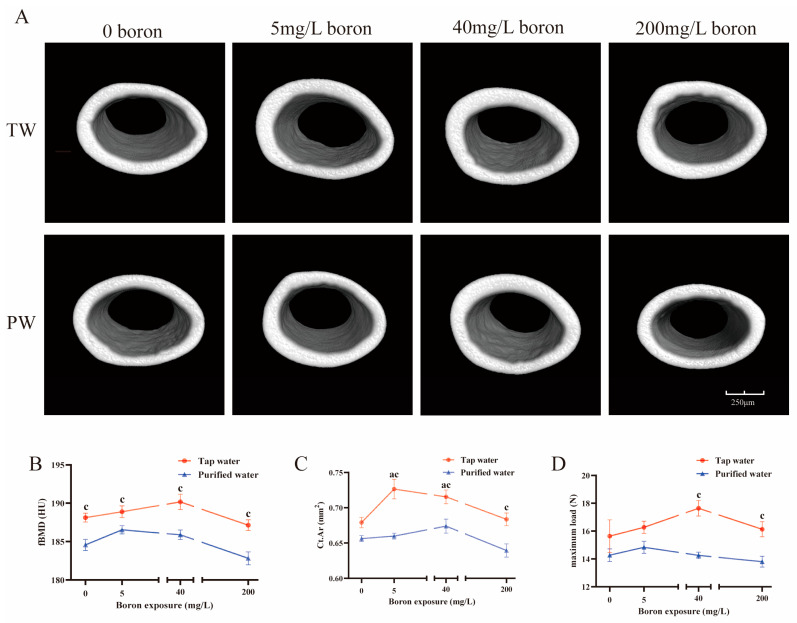
Bone microstructure and biomechanical parameters of the femora were analyzed by a three-dimensional microcomputed tomography system. (**A**) The three-dimensional microcomputed tomography images of femoral bones by microcomputed tomography (micro-CT). (**B**) Femur bone mineral density. (**C**) The mean of cortical bone area. (**D**) Maximum load. The values are presented as means with the error bars indicating SEM; *n* = 7 mice/group. ^a^: there is a significant difference compared with the group drinking tap water without boron exposure (*p* < 0.05). ^c^: there is a significant difference between the two groups which had the same exposed dose of boron but combined with different water (*p* < 0.05).

**Figure 3 nutrients-16-02881-f003:**
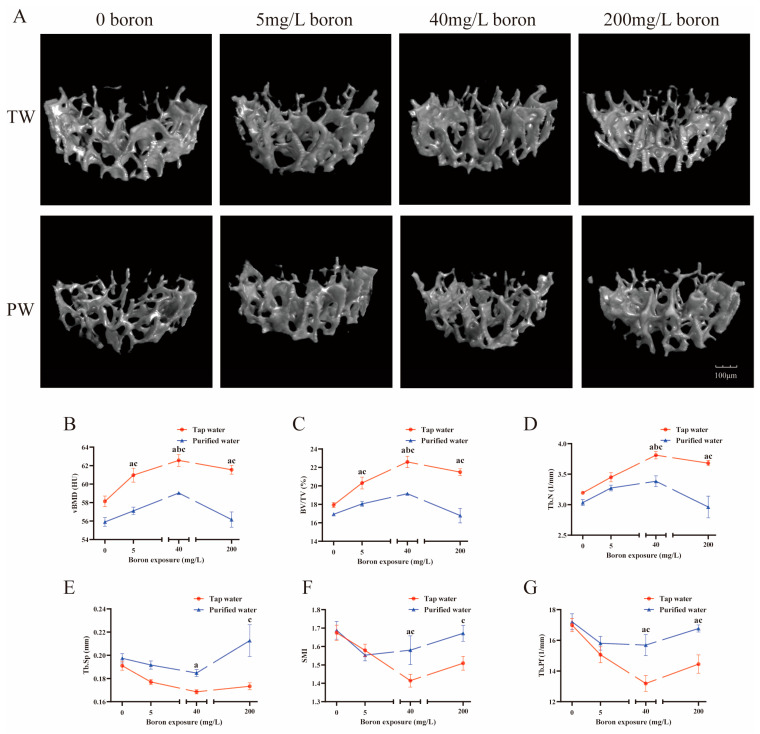
Bone microstructure parameters of the vertebrae were analyzed by a three-dimensional microcomputed tomography system. (**A**) The three-dimensional microcomputed tomography images of vertebrae by microcomputed tomography (micro-CT). (**B**) Vertebrae bone mineral density (**C**) Bone volume/total volume. (**D**) Trabecular bone number. (**E**) Trabecular separation. (**F**) Structure model index. (**G**) Trabecular pattern factor. The values are presented as means with the error bars indicating SEM; *n* = 6 mice/group. ^a^: there is a significant difference compared with the group drinking tap water without boron exposure (*p* < 0.05). ^b^: there is a significant difference compared with the group drinking purified water without boron exposure (*p* < 0.05). ^c^: there is a significant difference between the two groups which had the same exposed dose of boron but combined with different water (*p* < 0.05).

**Figure 4 nutrients-16-02881-f004:**
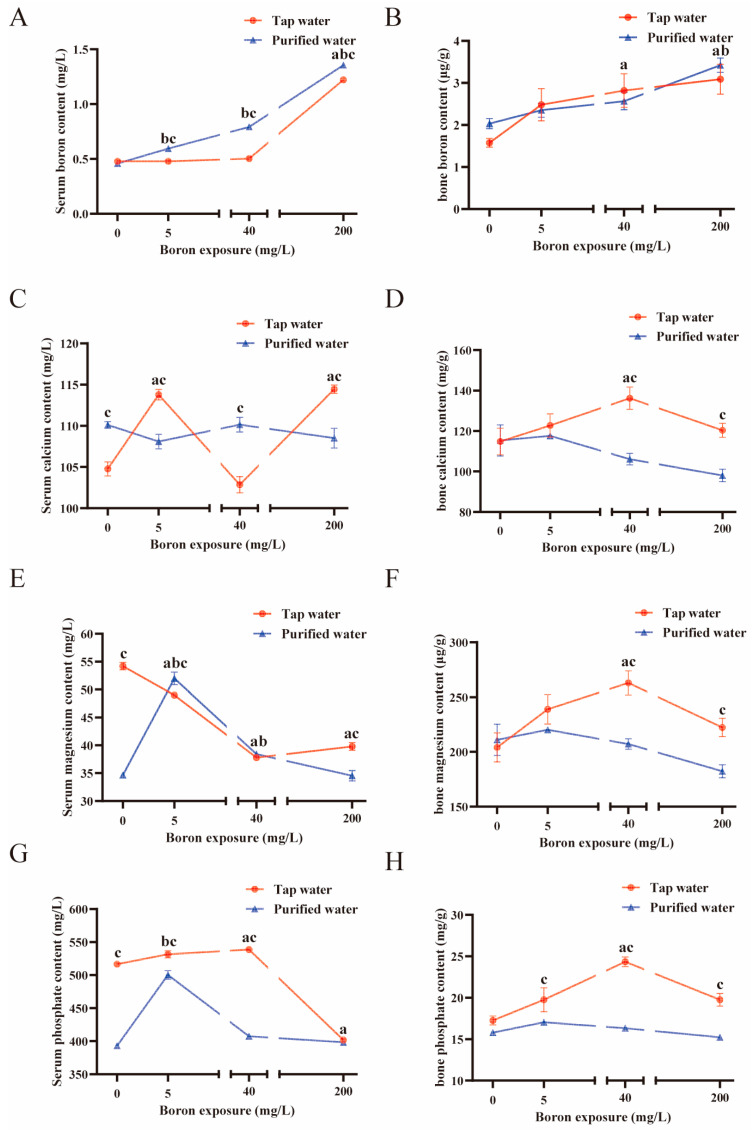
Blood and bone minerals in the mice drinking two types of water with boron. (**A**) The serum boron content. (**B**) The bone boron content. (**C**) The serum calcium content. (**D**) The bone calcium content. (**E**) The serum magnesium content. (**F**) The bone magnesium content. (**G**) The serum phosphate content. (**H**) The bone phosphate content. The values are presented as means with the error bars indicating SEM; bone: *n* = 8 mice/group; serum: *n* = 3 mice/group. ^a^: there is a significant difference compared with the group drinking tap water without boron exposure (*p* < 0.05). ^b^: there is a significant difference compared with the group drinking purified water without boron exposure (*p* < 0.05). ^c^: there is a significant difference between the two groups which had the same exposed dose of boron but combined with different water (*p* < 0.05).

## Data Availability

The original contributions presented in the study are included in the article/Appendix A, further inquiries can be directed to the corresponding authors.

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
