# Peer review of "Low-Mineral Water Diminishes the Bone Benefits of Boron"

_nutrients, 2024, doi:10.3390/nu16172881_

Round 1

Reviewer 1 Report

Comments and Suggestions for Authors

Please find my comments within the file attached.

Comments on the Quality of English Language

Minor editing of English language required.

Author Response

General comments.

Improve the English language.

Carefully revise upper and lowercase formatting, mostly in figures captions and Graph axis.

Response: Thanks for pointing out the problem and we have checked and corrected grammar and spelling errors throughout the manuscript.

Specific comments.

Introduction.

Comment 1: In what regards published articles mentioned in the introduction section, from line 60 to line 71. Please clarify, whenever it is not evident, if the impact of boron was evaluated in an adequacy setting of other minerals (namely calcium, magnesium and phosphorous) or not.

Response 1: Thanks for reminding us. Some of these studies have reported varying effects of boron in the context of both inadequate and adequate mineral (such as calcium, magnesium, and phosphorus) intake, as well as the impact of additional mineral supplementation based on adequate intake levels of minerals. We rewrite this section to clarify this (introduction, line 71-81).

Results.

Comment 2: “Within the groups that drank purified water, the serum OPG/RANKL was higher in the group exposed to 5 mg/L boron (P<0.05, Figure 1D).” Should be “Within the groups that drank purified water, serum OPG/RANKL levels were the highest in the group exposed to 5 mg/L boron (P<0.05, Figure 1D).”

Please check the entire paragraph for similar needed corrections/adjustments.

Response 2: Thank you for your advice. We rewrote the “Results”. In groups consuming the same water, when only one dose of boron exposure is found to be significant, we utilized the terms "highest" or "lowest" in the description. (Results, Line 225, 226, 229, and 230)

Comment 3:Similarly to what was abovementioned, the following sentence needs to be adjusted, as OCN levels are significantly lower for both 5 and 200 mg/L boron.                                                                                 

“BALP increased in the group exposed to 5 mg/L boron but decreased in the group exposed to 200 mg/L boron, which also had lower OCN levels (P<0.05, Figure 1A and 1B).”

Response 3: We now rewrote this sentence for clarity. “BALP increased in the group exposed to 5 mg/L boron but decreased in the group exposed to 200 mg/L boron (P<0.05, Figure 1A). OCN decreased in the group exposed to 5 mg/L and 200 mg/L boron (P<0.05, Figure 1B)”. (Results, line 236-238)

Comment 4:Figure 1 caption. There is no need to systematically repeat serum, as it is mentioned in the main/first sentence.

Response 4: Thanks for your careful work. We have deleted the repeated “serum” in the Figure 1 caption. (Figure 1 caption).

Comment 5: “Mice drinking purified water, compared to those drinking tap water, presented lower vertebra and femur BMD, either with or without boron co-exposure (P<0.05, Figure 2H and 3C).” Should be “Mice drinking purified water, compared to those drinking tap water, presented lower vertebra (Figure 2H) and femur (Figure 3C) BMD, either with or without boron co-exposure (P<0.05).” Adjust accordingly in the entire results section.

Response 5: We agree with your suggestion. The description is unclear. Now the figure was directly cited following the description of the corresponding indicator. (Results, line 215-219, line 232, line 239, line 239-240, line 247-260, line 282-283)

Comment 6: “Serum calcium increased when co-exposed to 0 mg/L and 40 mg/L boron but decreased when co-exposed to 5 mg/L and 200 mg/L boron in the mice drinking purified water (P<0.05, Figure 4C).” By observing Figure 4C, it can be seen that serum calcium levels are constant for purified water animals, independently of boron concentration. And this fact is not perceived from your results description.

Response 6: We thank the reviewer for bringing to our attention the fact that our description in this sentence was not sufficiently clear. The finding presented here was the difference between the groups drinking different water. We revised this sentence in the manuscript to enhance its clarity. “The serum magnesium in mice drinking purified water decreased when co-exposed to 0 mg/L and 200 mg/L boron but increased when co-exposed to 5 mg/L boron (P<0.05, Figure 4E), and serum calcium increased when co-exposed to 0 mg/L and 40 mg/L boron but decreased when co-exposed to 5 mg/L and 200 mg/L boron (P<0.05, Figure 4C), compared to those in mice drinking tap water.” (Results, line 271-275)

Comment 7:Please, check the statistical result for 40 mg/L in what regards this sentence: Serum boron was higher in the group exposed to 40 mg/L and 200 mg/L boron (P<0.05, Figure 4A). It does not seem correct by looking into the Figure 4A in what regards 40mg/L for tap water.

Response 7: Again, we thank you for your careful reading. You are right. In mice consuming tap water, exposure to 40 mg/L boron did not result in a statistically significant difference in serum boron compared to the control group with no boron exposure. It has been corrected. “Serum boron was higher in the group exposed to 200 mg/L boron (P<0.05, Figure 4A).” (Result, line 284)

Discussion.

Comment 8:Within boron groups there was no statistical differences regarding CTX levels when comparing both waters. So, adjust the following sentence: “However, only the highest boron exposure significantly promoted bone resorption while combined with purified water, compared with tap water.”

Response 8: Thanks for pointing out this mistake. It did not compare with mice drinking tap water here. “The highest boron exposure (200 mg/L) significantly promoted bone resorption in mice consuming purified water.” (Discussion, line 351-353)

Comment 9:The impact is quite tenuous upon TRACP by boron with tap water. So, adjust the following sentence: “There was a similar trend but not significant, when combined with tap water”.

Response 9: Thanks to the reviewer for pointing it out. When we re-analyzed the data according to the suggestion of reviewer 2, TRACP increasing in mice drinking tap water was significant when exposed to 40 mg/L and 200 mg/L boron (Result, line 238-239). Thus, “osteoclast activity exhibited an increase as boron exposure at 40 mg/L and 200 mg/L either drinking tap water or purified water, but only decrease in group drinking purified water when exposed to 5mg/L boron.” (Discussion, line 356-359)

Comment 10:The following sentence relates to BALP or to all the previously - within the discussion section - mentioned parameters? “When combined with purified water, boron exhibits a stronger promoting effect at lower concentrations and a weaker inhibitory effect at higher concentrations.” 

The sentence should be adjusted if it is referring just to BALP. As can be observed in Figure 1A a) no differences are seen between boron-treated groups for both waters and b) 200 mg/L boron-purified water impact upon BALP is not different from 0 mg/L boron-purified water impact. So, boron shows a promoting effect at lower concentrations and no effect at the highest boron concentration.

Response 10: We agree with this comment. “Osteoblast activity initially increased at 5 mg/L boron exposure and decreased as boron exposure increased (at 200 mg/L) in mice drinking tap water. When combined with purified water, boron promoted osteoblast activity at lower concentrations (5 mg/L). The effect weakened at higher boron exposure (40 mg/L) and demolished at highest boron exposure (200 mg/L).” (Discussion, line 367-371)

Comment 11:Improve English language. “They only release into the blood when bone resorption.”

Response 11: We rewrited this sentence. “The majority of OCN produced by osteoblasts is incorporated into the bone matrix and only enters the bloodstream when bone resorption.” (Discussion, line 372-373)

Comment 12:Correct Figure 1B lettering or the following sentence as they do not match. A significant difference is pointed out in Figure 1B for 5 mg/L boron. “Conversely, when combined with tap water, the impact of boron exposure on OCN was not significant until the highest dose, at which point a significant decrease was observed.”

Response 12: Thank you again for your careful work. “Mice drinking purified water had lower serum OCN than mice drinking tap water without boron exposure, but its serum OCN increased and higher than that in mice drinking tap water when exposed to boron. Besides, the serum OCN in mice drinking tap water decreased when exposed to boron (at 5 mg/L and 200 mg/L)”. (Discussion, line 375-378)

Comment 13:Differences were not always significant for all boron-concentrations. Adjust. “In the groups drinking purified water, the positive indicators of bone structure, including BV/TV, Tb.Th, Tb.N of the vertebrae, and Tt.Ar of the femora were all lower compared to the groups drinking tap water group. Conversely, the negative indicators, such as Tb.Sp, Tb.Pf, and SMI of the vertebrae, were higher than those observed in the groups drinking tap water group. These findings suggest that exposure to pure water adversely affects bone structure, irrespective of the presence of boron [27].”

Response 13: We deleted these sentences and concentrated on the analysis of the interaction between drinking water type and boron exposure. (Discussion, line 394-399)

Comment 14:What is low boron-dose? Regarding the sentence: “Briefly, low-dose boron exposure demonstrated beneficial effects on bone structure by enhancing the BV/TV and Tb.N, while concurrently reducing Tb.Pf, and SMI of the vertebrae.” Example: no significant differences are shown for 5 mg/L boron between the water groups for Tb.N and Tb.Pf.

Response 14: Thanks for reminding us. We rewrote these sentences and added the dose of boron exposure. “In mice drinking tap water, low-dose boron exposure (5 mg/L or 40 mg/L) demonstrated beneficial effects on bone structure by enhancing the vertebral BV/TV, Tb.N, and femoral Tt.Ar, while concurrently reducing Tb.Pf, Tb.Pf, and SMI of the vertebrae. The advanta-geous effects on vertebral Tb.Pf, SMI, and femoral Tt.Ar were attenuated as the boron ex-posure increased (200 mg/L). All of these benefits of boron were negated at some exposure doses in mice drinking purified water.” (Discussion, line 394-399)

Comment 15:Are these sentences previous results? Adjust. 

  1. a) “The boron content in the under-mineralized bone was significantly lower compared to the fully mineralized bone [28].”
  2. b) “Serum magnesium levels varied in response to boron's regulation of bone magnesium in mice drinking tap water. Although consuming purified water resulted in a decrease in serum magnesium [26,27].”

Response 15: a) cited other’s study (Discussion, line 417-418) and b) cited our previous studies. We have removed sentence b) to avoid confusion.

Comment 16: “The groups drinking purified water only exhibited a significant increase in bone boron content at high doses of boron exposure.” At the highest dose, as can been seen in Figure 4B. Adjust.

Response 16: Thanks for pointing this out. We have changed the sentence to: “Bone boron only increased when exposed to 200 mg/L boron.” (Discussion, line 420)

Comment 17:As mentioned above boron with purified water had no impact upon serum calcium content. Adjust. “Combination with purified water not only diminished the promoting effect of boron on bone calcium, but also weakened boron's promoting effect on blood calcium.”

Response 17: Thank you for your reminder. “Drinking purified water diminished the promoting effect of boron on bone and serum calcium.” (Discussion, line 428-429)

Comment 18:There is no significant difference in Figure 4F with b lettering. So adjust the following sentence. “Bone magnesium in mice drinking purified water group did change as boron exposure.”

Response 18: This was our mistake. This should be “did not”. We have corrected it. (Discussion, line 430-431)

Comment 19:What is the ratio of calcium to magnesium in diet+water of each water group? Calcium and magnesium can interfere with each other and the optimum ratio value is around 2. Several quite recent articles discuss this issue. Please include and discuss. Then adjust the entire section of the discussion related to magnesium, calcium and boron effects.

Response 19: Thank you for putting forward this important question. The ratio of calcium to magnesium intake is important for bone due to their competitive absorption. “In this study, the ratio of calcium to magnesium in diet+water was 3.49 in mice drinking tap water and 3.48 in mice drinking purified water. (Results, line 205; Table S1) These values are close to the range of 2.2-3.2, which was identified as the most protective for bone health in Hibler's study. Additionally, there were no significant differences in these ratios between groups consuming different types of water. This suggests that the observed differences in bone and mineral metabolism, microstructure, and biomechanics between the groups drinking different water may not be attributed to the calcium-to-magnesium intake ratio.” (Discussion, line 341-347)

Reviewer 2 Report

Comments and Suggestions for Authors

I have received for review the revised manuscript entitled "Low-mineral water diminishes the bone benefits of boron". In my opinion, the manuscript needs significant revision an cannot be recommended for publication before corrections.

The main issue with the experimental design is the statistical approach, which uses a low-power test – LSD. It is not recommended to use the LSD test in preclinical studies, as it does not correct for multiple comparisons. In this study, a two-way ANOVA with boron supplementation and water type as the main factors should be the first choice of statistical model. The number of animals used in the assays is rather low (from 4 to 7). The low number of experimental animals, combined with a post hoc test that does not correct for multiple comparisons, results in a high type I error and low reliability. Was a power analysis performed?

I will not review the discussion section, which depends on the results of the statistical analysis that, in my opinion, is currently not appropriate for this type of study.

Specific comments for the introduction and materials and methods sections, as well as the presentation of the results:

L39: In a scientific paper, please use the proper name “the Persian Gulf region” instead of “the Gulf region.”

L55: Reference to that previous study is needed.

The introduction requires substantial rewriting. The sections discussing the effects of boron on bones, bone cells, and its synergistic interactions with other minerals are scattered throughout the introduction. For clarity and coherence, these elements should be organized into a more structured format.

Additionally, the introduction would benefit from a clearer distinction between the background information and the study's objectives. The aim of the study is not clearly stated. It should be elaborated and based on a clearly defined research hypothesis. Also, correct to “BALB/c mice.”

L108: The year “2022” is missing.

The description of the experimental layout is unclear and needs significant improvement.

The reported weights of the mice (16.7g ±11.2 g) are confusing. The division of mice into eight groups and their respective treatments is not clearly outlined. The statement "boron: No Decision" for purified water is ambiguous.

L127: Unclear. Were only left femora harvested? Were all the assays (uCT, mechanical testing, mineral content analysis) performed on the same bone?

L129: What was the volume of sampled blood?

L143: Is this a proper reference?

L151: What were the conditions under which the bones were secured during the uCT analysis to prevent them from drying? What software was used for bone microarchitecture evaluation?

The description of the bone mechanical analysis must be provided in more detail, especially as cited reference [27] describes the protocol for rat bone testing, and smaller mice were used in this study. How were measurements from the femoral head to the most distal point of the medial condyle used for calculating a rather simple parameter such as maximal load, which can be directly read from load-deflection curves?

Figures: The data should be presented in bar plots instead of line graphs. Line graphs are generally used for data like time-series plots where there is continuity between measurement time points. For data with discrete different doses of an experimental factor (boron), such a relationship cannot be assumed, and using line connections between points also suggests the existence of some.

Fig 1: Why were only 4 mice per treatment used?

Fig 2B: What is the unit of Tr.Ar?

Fig 4: Serum boron content – no SEM?Fig 1. Why only 4 mice er treatment were used?

Fig 2B – what is the unit of Tr.Ar ?

Fig 4 – serum boron content – no SEM ?

Comments on the Quality of English Language

There are several grammatical errors, awkward phrasings, and instances where the language can be more precise and concise.

Author Response

Comment 1: The main issue with the experimental design is the statistical approach, which uses a low-power test – LSD. It is not recommended to use the LSD test in preclinical studies, as it does not correct for multiple comparisons. In this study, a two-way ANOVA with boron supplementation and water type as the main factors should be the first choice of statistical model. The number of animals used in the assays is rather low (from 4 to 7). The low number of experimental animals, combined with a post hoc test that does not correct for multiple comparisons, results in a high type I error and low reliability. Was a power analysis performed? I will not review the discussion section, which depends on the results of the statistical analysis that, in my opinion, is currently not appropriate for this type of study.

Response 1: We thank the reviewer for bringing up this point. We agree with this comment. The likelihood of committing a Type I error increases with the number of group comparisons when using the LSD method. And there was two factors, water type and boron exposure. Accordingly, we re-analyzed the statistical significance of the data. Two-way ANOVA procedure in general linear models was used to analyze interaction effect of two factors, water type (tap water vs purified water) and boron exposure level (0, 5, 40, and 200). Bonferroni were used to examine both main effects and simple effects. (Materials and Methods, line 191-195)

Specific comments for the introduction and materials and methods sections, as well as the presentation of the results:

Comment 2:L39: In a scientific paper, please use the proper name “the Persian Gulf region” instead of “the Gulf region.”

Response 2: Thanks for pointing this out. We have corrected it. (Introduction, line 39)

Comment 3:L55: Reference to that previous study is needed.

Response 3: These are the preliminary data from our ongoing prospective cohort studies. The study is still in progress, and the results have not yet been published.

Comment 4:The introduction requires substantial rewriting. The sections discussing the effects of boron on bones, bone cells, and its synergistic interactions with other minerals are scattered throughout the introduction. For clarity and coherence, these elements should be organized into a more structured format.

Response 4: Thank you for your proposal. This part of the introduction in our original manuscript lacked clarity. It has been rewritten accordingly. (Introduction, line 62-81)

Comment 5:Additionally, the introduction would benefit from a clearer distinction between the background information and the study's objectives. The aim of the study is not clearly stated. It should be elaborated and based on a clearly defined research hypothesis. Also, correct to “BALB/c mice.”

Response 5: Thank you very much for reminding us. We modified the final section and corrected the mistake. (Introduction, line 106-109)

Comment 6:L108: The year “2022” is missing.

Response 6: Thank you. We added it in text. (Materials and Methods, line 114)

Comment 7:The description of the experimental layout is unclear and needs significant improvement.

Response 7: Thank you for your very helpful suggestions to improve our paper. We have reorganized the "Materials and Methods" section and added detailed descriptions of certain methods to enhance clarity. (Materials and Methods, line 144-180)

Comment 8:The reported weights of the mice (16.7g ±11.2 g) are confusing. The division of mice into eight groups and their respective treatments is not clearly outlined. The statement "boron: No Decision" for purified water is ambiguous.

Response 8: Thanks again for this careful observation. It was our mistake. The weights of the mice were 16.7 g ±2.2 g. (Materials and Methods, line 119) The “No Decision” means “below the detection limit”. (Materials and Methods, line 124)

Comment 9:L127: Unclear. Were only left femora harvested? Were all the assays (uCT, mechanical testing, mineral content analysis) performed on the same bone?

Response 9: Thanks for raising this point. It was a problem in our writing. Both the left and right femora were harvested and utilized for distinct experimental tests. left femora were harvested for the evaluation of bone microstructure. After scanning, the mineral content of the left femora was evaluated. Right femora were assessed for biomechanical properties. (Materials and Methods, line 134-137)

Comment 10:L129: What was the volume of sampled blood?

Response 10: One mL. (Materials and Methods, line 137)

Comment 11:L143: Is this a proper reference?

Response 11: Thank you for this reminder. We have removed it.

Comment 12:L151: What were the conditions under which the bones were secured during the uCT analysis to prevent them from drying? What software was used for bone microarchitecture evaluation?

Response 12: It was wrapped with normal saline-soaked gauze kept hydrated during the uCT analysis. Following scanning, NRecon (version 1.7.4.6, Bruker microCT, Kontich, Belgium) was used for data reconstruction, and Data Viewer (version 1.5.6.2, Bruker microCT, Kontich, Belgium) adjusted the angles of coronal, sagittal, and horizontal images. Then Skyscan CT-Analyser (version 1.23.0.2, Brucker AG, Billerica, Massachusetts, USA) is used to calculate bone morphological parameters within the regions of interest (ROI), which were defined as starting from 0.2 mm to 0.6 mm below the lumbar vertebra growth plate and 3.5 mm to 5.5 mm above the femoral growth plate, respectively (Supplemental Figure S1). The parameters including vertebral bone volume/total volume (BV/TV), trabecular bone number (Tb.N), trabecular separation (Tb.Sp), trabecular pattern factor (Tb.pf), structure model index (SMI), and bone mineral density (BMD), femoral BMD and mean of cortical bone area (Ct.Ar) were evaluated under the guidelines outlined by the American Society for Bone and Mineral Research (ASBMR) Histomorphometry Nomenclature Committee. CTvox (version 3.3.1, Bruker microCT, Kontich, Belgium) is employed for three-dimensional reconstruction analysis of the femur. (Materials and Methods, line 161-175)

Comment 13:The description of the bone mechanical analysis must be provided in more detail, especially as cited reference [27] describes the protocol for rat bone testing, and smaller mice were used in this study. How were measurements from the femoral head to the most distal point of the medial condyle used for calculating a rather simple parameter such as maximal load, which can be directly read from load-deflection curves?

Response 13: Again, we thank the reviewer for her careful reading. Yes, the length and strength of the femora exhibit significant differences between rats and mice. Consequently, we adjusted the span of the supporting bars and the speed of tensile loading accordingly. Briefly, the bone was immersed in a physiological saline solution to maintain its hydration before mechanical testing. Subsequently, the bone was positioned horizontally on two supporting bars with a span of 10 mm, and a central loading force was applied at the mid-diaphysis. The femur was subjected to tensile loading at a rate of 1 mm/s until fracture. The load-deformation curve was then recorded using a universal material testing machine (Instron1011, Instron Corporation, USA). (Materials and Methods, line 183-188)

The maximum load can be obtained from the load-deformation curve directly.

Comment 14:Figures: The data should be presented in bar plots instead of line graphs. Line graphs are generally used for data like time-series plots where there is continuity between measurement time points. For data with discrete different doses of an experimental factor (boron), such a relationship cannot be assumed, and using line connections between points also suggests the existence of some.

Response 14: Thank you for your insightful comments. The purpose of presenting the data in line graphs was to illustrate the interaction effects of water type and boron exposure. Consequently, these figures were not modified.

Comment 15:Fig 1: Why were only 4 mice per treatment used?

Response 15: Thank you for your detailed review of our figures. This is our serious mistake. Eight samples per group, rather than four, were utilized for serological testing. We corrected the Figure 1 legend. (Figure 1 legend)

After serological testing, only three leftover serum samples exceeded 100 µL per group. Therefore, the serum mineral analysis was conducted using only three samples per group. (figure 4 legend)

Comment 16:Fig 2B: What is the unit of Tr.Ar?

Response 16: Thanks again for your detailed review of our figures. The unit in our model was mm. Thus, It was “mm2” here. In Figure 3, we have replaced "mean of total cross-sectional tissue area of cortical bone (Tt.Ar)" with "mean of cortical bone area (Ct.Ar)" because Ct.Ar more directly reflects the structure of cortical bone. (Figure 3)

Comment 17:Fig 4: Serum boron content – no SEM?

Response 17: SEM has been marked. Due to the substantial disparity between the groups, the relatively minor SEM is invisible. (Figure 4)

Comment 18:Comments on the Quality of English Language

There are several grammatical errors, awkward phrasings, and instances where the language can be more precise and concise.

Response 18: Thank you very much for your criticism. We have checked and corrected grammar and spelling errors throughout the manuscript.

Reviewer 3 Report

Comments and Suggestions for Authors

Thank you for your contribution to this journal. It is very interesting animal study results, even though several clinical aspects should be considered. 

The author have provided boron's role in bone health and metabolism with other minerals such as calcium, magnesium and phosphorus.

As the authors declared, this study was done on animal based, not human.

Therefore, there are many other contributing factors such as other mineral intake by food, supplements, and medical conditions, etc., which is a big limitation to apply human bone health. 

In area of the purified water is needed, some are not consider on bone health in some aspect, because water in more important for life consistency, even though bone health is also important for a long period accumulation effect. 

This research result itself is very informative, well designed,  and very clear result presentation, even though this study is animal based.

So, I have no other comments to revise, however, further study, as the authors mentioned, of ecological study is needed, which is very difficult as ethical problem rise. 

Author Response

Thank you for your contribution to this journal. It is very interesting animal study results, even though several clinical aspects should be considered. 

The author have provided boron's role in bone health and metabolism with other minerals such as calcium, magnesium and phosphorus.

As the authors declared, this study was done on animal based, not human.

Therefore, there are many other contributing factors such as other mineral intake by food, supplements, and medical conditions, etc., which is a big limitation to apply human bone health. 

In area of the purified water is needed, some are not consider on bone health in some aspect, because water in more important for life consistency, even though bone health is also important for a long period accumulation effect. 

This research result itself is very informative, well designed,  and very clear result presentation, even though this study is animal based.

So, I have no other comments to revise, however, further study, as the authors mentioned, of ecological study is needed, which is very difficult as ethical problem rise. 

Response: We appreciate the reviewer’s appraisal of the importance of our investigation. Thank you for your affirmative evaluation of our manuscript.

Round 2

Reviewer 2 Report

Comments and Suggestions for Authors

Thank you for addressing the issues that I highlighted.